# Comparison of End-to-End Neural Network Architectures and Data Augmentation Methods for Automatic Infant Motility Assessment Using Wearable Sensors

**DOI:** 10.3390/s23073773

**Published:** 2023-04-06

**Authors:** Manu Airaksinen, Sampsa Vanhatalo, Okko Räsänen

**Affiliations:** 1BABA Center, Pediatric Research Center, Children’s Hospital, Helsinki University Hospital and University of Helsinki, 00290 Helsinki, Finland; 2Unit of Computing Sciences, Tampere University, 33720 Tampere, Finland; sampsa.vanhatalo@helsinki.fi (S.V.);

**Keywords:** human activity recognition, classifier architectures, wearable technology, infant motility

## Abstract

Infant motility assessment using intelligent wearables is a promising new approach for assessment of infant neurophysiological development, and where efficient signal analysis plays a central role. This study investigates the use of different end-to-end neural network architectures for processing infant motility data from wearable sensors. We focus on the performance and computational burden of alternative sensor encoder and time series modeling modules and their combinations. In addition, we explore the benefits of data augmentation methods in ideal and nonideal recording conditions. The experiments are conducted using a dataset of multisensor movement recordings from 7-month-old infants, as captured by a recently proposed smart jumpsuit for infant motility assessment. Our results indicate that the choice of the encoder module has a major impact on classifier performance. For sensor encoders, the best performance was obtained with parallel two-dimensional convolutions for intrasensor channel fusion with shared weights for all sensors. The results also indicate that a relatively compact feature representation is obtainable for within-sensor feature extraction without a drastic loss to classifier performance. Comparison of time series models revealed that feedforward dilated convolutions with residual and skip connections outperformed all recurrent neural network (RNN)-based models in performance, training time, and training stability. The experiments also indicate that data augmentation improves model robustness in simulated packet loss or sensor dropout scenarios. In particular, signal- and sensor-dropout-based augmentation strategies provided considerable boosts to performance without negatively affecting the baseline performance. Overall, the results provide tangible suggestions on how to optimize end-to-end neural network training for multichannel movement sensor data.

## 1. Introduction

Developments in wearable electronics, and especially in integrated sensors with inertial measurement units (IMUs), have made automatic human activity recognition (HAR) feasible in various use scenarios. Typical HAR systems aim at detecting various everyday activity patterns of a person (e.g., sitting, walking, running). These data can then be used for various purposes ranging from personal activity logs [1] to medical analyses [2,3] (see [4] for a review). One particularly promising example of the latter HAR application area involves automatic monitoring of infant movements using wearable intelligent garments [5,6,7,8]. As the observed motor behavior of a child is closely coupled to the development and integrity of the child’s central nervous system, abnormal movement patterns or lack of certain movement skills at a given age can be predictive of deviant neurocognitive development such as cerebral palsy [9]. In the same manner, automatic at-home monitoring of an infant undergoing a therapeutic intervention can produce data on intervention efficacy that is difficult to obtain during time-limited (and often uncomfortable) hospital visits [10].

In our previous work [11,12], we developed comfortable-to-wear infant overalls (or jumpsuit) called “Maiju” (Figure 1; Motor Assessment of Infants with a JUmpsuit) with integrated IMUs in each limb for automatic assessment of infant movement repertoire for out-of-the-lab settings. We also collected a unique dataset consisting of visually annotated posture and movement data from several infants. By measuring movements of a child using Maiju over some tens of minutes of independent movement, followed by a neural-network-based signal analysis pipeline, our setup was able to recognize different postures and movement patterns of the child with human-equivalent accuracy. However, the signal classification pipeline and its training protocol used in the initial study were not results of systematic optimization, but rather an attempt to construct a functional solution for the task. Despite the apparent success, many technical details remain to be defined more systematically to meet our ultimate goal to develop a widely scalable system for out-of-hospital movement analysis in infants.

In the previous study [11], classification in the movement track turned out to be more challenging compared to the posture track (≈94% unweighted average F1-score for posture, ≈73% for movement). Similar results for posture classification were also reported in [5]. Furthermore, comparable posture classification results were obtained with a traditional feature-based support vector machine (SVM) classifier compared to the end-to-end convolutional neural network (CNN) classifier, but the CNN produced statistically significant improvements for movement classification. Based on this, only the movement track-based dataset was selected for the experiments within the present study.

In this work, we take a more principled approach to the technical development of a infant movement classifier from wearable IMUs. More specifically, we investigate different end-to-end neural network architectures for multisensor IMU data processing in infant movement analysis. We also analyze the utility of different training data augmentation strategies for classifier performance and robustness. Different system variants are tested in conditions that simulate issues encountered in practical out-of-the-lab use scenarios, such as sensor dropout or packet loss due to real-time wireless data transfer issues. We also compare performance of different system architectures to their memory footprint and computational complexity. As a result, we find many such strategies in sensor signal encoding and time series modeling that may benefit also many other HAR applications that are based on multisensor setups.

## 2. Background

The Maiju jumpsuit reported in [11] records posture and movement patterns of infants by measuring their movements with a set of limb-mounted IMUs (see Figure 1) and by transforming the resulting signals into posture/movement categories using an end-to-end neural network processing pipeline. Our initial study showcasing the smart jumpsuit contained a broad scope, detailing the process of textile design, recording setup, dataset design including annotation and its analysis, and, finally, the evaluation of automatic classification performance. As such, a broad comparison and optimization of varying classifier architectures was beyond the scope of the study. Given the real-world relevance of the analysis task and uniqueness of the collected dataset, the question of the effects of varying end-to-end classifier architectures is poorly understood and requires proper investigation.

The general neural network architecture presented in [11] can be thought of as consisting of an encoder module that performs intra- and intersensor channel fusion to obtain a unified feature representation from the raw triaxial accelerometer and gyroscope signals of each IMU, and a time series modeling module for modeling the temporal context in the activity recognition process (see Figure 2). During training, both the encoder and time series module are jointly optimized for the classification task in an end-to-end manner.

The HAR literature contains multiple proposals to perform raw-input feature extraction for the encoder model (e.g., [4,13]), and there is a growing number of popular time series modeling methods making use of deep learning [14,15,16]. The first goal of the present study is to find the best overall combination of the two modules and also to understand the performance of different modules in movement classification using the previously collected dataset. Varying classifier architectures can have varying levels of complexity in terms of model size (number of trainable parameters, memory footprint) and computational complexity (model training time, generation time). The second objective of the present study is to map the tested classifiers within this three-dimensional (performance vs. size vs. speed) space.

Finally, the recording setup of Maiju performs real-time streaming of the raw data to the controlling device (a mobile phone) over Bluetooth. This means that the real-life recordings might suffer from packet loss and/or sensor dropout. It has been well documented that data augmentation methods can improve model robustness in image and audio processing applications [17,18] as well as in HAR applications [19]. Thus, the final goal of the present study is to explore the effect of data augmentation methods for the Maiju jumpsuit movement classification task in optimal conditions and with varying levels of packet loss or sensor dropout.

## 3. Data Collection Method

The “Maiju” smart jumpsuit is a full body garment that allows spontaneous unrestricted movements of approximately 4–16-month-old babies. The jumpsuit performs triaxial accelerometer and gyroscope recordings at a 52 Hz sampling rate from four Suunto Movesense (www.movesense.com, accessed 6 March 2023) sensors with proximal limb placements. The accelerometer measures linear acceleration in m/s${}^{2}$ (range ±8 g) and the gyroscope measures angular velocity in deg/s (range ±500 deg/s). The Movesense sensors are 36.6 mm in diameter and 10.6 mm in thickness, weigh 30 g, and are waterproof and removable from programmable mounts. The sensors stream the raw data via Bluetooth 4.0 Low Energy to a controlling smartphone (iPhone 8; see Figure 1b).

The dataset collected in our previous study [11] was used for the experiments within the present study. The dataset contains recordings of independent nonassisted movement of 22 infants from approximately 30 min play sessions facilitated and monitored by a physiotherapist. The infants were approximately 7 months old and normally developing, meaning that their approximate motility skills ranged from turning to crawling (see Table 1 for further details). All subjects gave their informed consent for inclusion before they participated in the study. The study was conducted in accordance with the Declaration of Helsinki, and the protocol was approved by the Ethics Committee of Childrens Hospital, Helsinki University Hospital (project identification code: VAURAS).

Each recording of the dataset was video-recorded and annotated by three independent experts based on a coding scheme reported in [11]. The annotations contain two parallel tracks: one for posture (five categories) and one for movement (seven categories; see Table 1 for a list).

In order to derive training class labels for the dataset, the previous study used the three parallel annotations for each recording within the so-called iterative annotation refinement (IAR) process to obtain the most likely one-hot labels for the movement categories [11]. In IAR, the probabilistic interpretation of the human annotations is iteratively weighted with the softmax output of the automatic classifier judgments, which results in more consistent decision boundaries between categories. In this study, we will use the previously-derived IAR-based labels for the movement classes of the dataset. For further details, see [11] and its supplementary material.

## 4. Methods

### 4.1. Data Preprocessing and Windowing

The data preprocessing and windowing process is illustrated in Figure 1b. First, the raw data from the jumpsuit sensors (4 IMUs, 3 accelerometer + 3 gyroscope channels each) are streamed into the recording device in packets of four subsequent time samples (4 timestamps + 4 × 6 channels). Due to sampling jitter, the timestamps and signals from each sensor are first linearly interpolated from the recorded timebase into a synthesized ideal timebase with a 52 Hz sampling rate that is uniform for each sensor. The interpolated signals are organized into a matrix with columns corresponding to different recording channels. The gyroscope bias is estimated (and subtracted) from each individual channel by computing the mean of the 64-sample segment with the smallest variance within the recording. Next, excess noise is removed from the recordings by median filtering each channel separately (5-sample median). Finally, the recording matrix of shape (24, $N_{rec}$), where $N_{rec}$ is the length of the recording in samples, is windowed into 120-sample-long (2.3 s) frames with 50% overlap using a rectangular window and stored into a tensor with shape ($N_{frames}$, 24, 120) to be used as an input to the classifier stage.

### 4.2. Baseline SVM Classifier

We utilize the SVM classifier reported in our previous study [11] as a performance baseline. The SVM classifier utilizes 14 basic features per channel, yielding a feature vector with a total dimensionality of 14 × (4 × 3 × 2) = 336 features per frame. The chosen features were signal mean, variance, max amplitude, min amplitude, signal magnitude area, energy, interquartile range, skewness, kurtosis, largest frequency component, weighted average frequency, frequency skewness, and frequency kurtosis of each channel. The multiclass SVM was trained as an ensemble system with the error-correcting output codes (ECOC) model utilizing linear kernel functions. The input vectors to the model were standardized with global mean and variance normalization. The evaluation was performed with leave-one-subject-out (LOSO) cross-validation.

### 4.3. Overall Classifier Design

The overall schematic of the processing pipeline is presented in Figure 2. For each recording, the preprocessed input tensor is fed into an encoder module that produces a feature representation of 160 bottleneck features from each frame. The bottleneck features are fed into the time series modeling module that tracks the frame-to-frame behavior of the bottleneck features, and finally produces the output movement classification for each frame. In our experiments, the encoder modules (Section 4.4) and time series modeling modules (Section 4.5) can consist of 5 distinct architectures, resulting in a total of 25 combinations to be investigated.

### 4.4. Compared Sensor Encoder Modules

#### 4.4.1. Dense Network

The “dense” encoder structure, presented in Figure 3a, is the most simple approach to learn an end-to-end feature representation. Each input frame is flattened across all the channels from all the IMUs into a vector of size 1 × 2880, and a fully-connected transformation with tanh activation is applied to it, reducing the dimensionality to 256. Three additional fully connected transformations follow, and the output is the 160-dimensional bottleneck feature vector for each frame. The inclusion of the “dense” encoder into the experiments is mainly motivated by the assumption that it acts as an anchor point for low-end performance compared to the more complex CNN architectures, as the number of trainable parameters is higher and the module does not utilize weight sharing.

#### 4.4.2. 1-D CNN

The 1-dimensional CNN system (Conv1D; Figure 3b) is a simple CNN system that utilizes spatial weight sharing over channels of filters to perform dimensionality reduction of the sensory input. The input tensor of the 1-D CNN system is treated so that the 24 channels are fully connected into the first convolution layer’s filter channels, and the convolution filters operate over the time axis of the windowed samples. Each convolutional layer has a filter size of 5 and stride of 2, effectually halving the temporal dimension of the input within each layer. The final layer of the 1-D CNN concatenates the obtained feature channels into a flat vector and performs a fully-connected transformation to obtain the output size of 160 features per frame. The activation function for each layer is a leaky rectified linear unit (LReLU).

#### 4.4.3. 2-D CNN with Individual Paths

The previous encoder systems (dense and 1-D CNN) work under the assumption that the same learned features can be extracted from the parallel accelerometer and gyroscope signals. In reality, the accelerometer and gyroscope sense complementary inertial information that potentially benefit from individualized features. The “2-D CNN with individual paths” (Conv2D-I; Figure 3c) splits the input paths of the encoder modules to perform feature extraction from the accelerometer and gyroscope separately. In addition, 2-D convolutions are utilized in the first two layers to perform spatially relevant channel fusion: the first layer fuses each sensors’ triaxial (xyz) channels into shared sensor-level channels, and the following layer fuses the four sensor-level channels of the measurement modality (i.e., acc or gyro) with the 2-D convolution. The accelerometer and gyroscope-specific features are concatenated, and two fully connected operations are performed to mix the domain-specific information to the final output representation. The activation functions for each layer are LReLUs.

#### 4.4.4. 2-D CNN with Individual and Shared Paths

The 2-D CNN with individual and shared paths (Conv2D-IS; Figure 3d) is identical to the 2-D CNN with individual paths, except that the encoder input has also an additional shared path, where shared filters are learned for the accelerometer and gyroscope alongside the individual paths. This encoder is equivalent to the encoder utilized in our initial study [11]. The addition of the shared path was reported to increase performance in a previous study regarding multi-IMU sensor HAR [13], and the comparison between the 2DCNN-I and 2DCNN-IS systems is interesting regarding the generalizability of the aforementioned findings.

#### 4.4.5. 2-D CNN with Sensor-Independent Modules

The 2-D CNN with sensor-independent modules (Conv2D-SI; Figure 3e) encoder aims to leverage a higher order of weight sharing compared to the previous CNN systems: the encoder is set to produce independent feature representations for each IMU sensor using a shared sensor model, whose outputs are finally concatenated. The sensor module utilizes similar architecture to the Conv2D-IS system (three total paths), with the exception that no intersensor mixing of channels is applied in any of the layers. As such, the encoder is forced to learn a representation that is generalizable for each sensor, regardless of its location in the recording setup.

### 4.5. Compared Time Series Modeling Modules

#### 4.5.1. Dense Network

In analog to its encoder counterpart, the “dense” time series modeling module (Figure 4a) acts as a baseline anchor point for the time series modules: The module has one hidden fully connected layer, and, in fact, does not perform any time series modeling but simply transforms the output of the sensor module into a movement classification decision.

#### 4.5.2. LSTM

Long short-term memory (LSTM [14]; Figure 4b) networks have become one of the most common recurrent neural network (RNN) architecture choices. LSTMs circumvent the vanishing gradient problem of RNNs by utilizing a distinct cell memory state that is manipulated (read/write/reset) by soft-gating operations. As such, LSTMs are ideal for learning long-term connections within the time series input data, such as those exhibited in speech and language. In the context of HAR, and especially within the utilized dataset, long-term connections are not probably overly emphasized, as infant movement is not preplanned over longer time periods. In addition, LSTM training can exhibit instability issues when training with smaller dataset sizes, which might make LSTM modeling nonideal for many medical applications where access to massive amounts of existing data may be limited, such as the present dataset.

#### 4.5.3. GRU

The gated recurrent unit (GRU [15]; Figure 4c) is a more recent variant of LSTM-inspired RNNs. Instead of utilizing distinct cell states, the GRU cell performs the learned soft-gated operations (read/write/reset) to the previous output state. This has the advantage of stabilizing the training compared to LSTMs, but with a trade-off in sensitivity to model long-term connections within the time series. As discussed in the LSTM section, the long-term connections are not expected to be overly important within the present dataset of infant motility.

#### 4.5.4. Bidirectional GRU

The bidirectional GRU (BGRU; Figure 4d) utilizes two GRU modules to perform the time series modeling: one in the forward direction, and another in the backward direction. The outputs of these two submodules are concatenated before the final fully-connected layer. The bidirectional design enables conditioning of the RNN time series decisions with the past and future information of the recording. This comes with the price of increased model complexity and the system being unable to perform causal inference for real-time applications.

#### 4.5.5. WaveNet

The WaveNet [16] architecture was first proposed as a speech synthesis specific modification of the ResNet [20] architecture used for image classification. The WaveNet consists of stacked blocks of gated dilated convolutions. The outputs of the blocks feed into the next block, and are also connected to the output of the block stack with residual connections. For time series modeling, these properties have multiple desirable features: first, the residual connections between the blocks enable stacking of multiple convolutional layers without the vanishing gradient problem, because during training the error gradient signal is able to flow freely along these shortcuts to the deeper layers; second, the stacked dilated convolutions reach a sufficiently wide receptive field of frames that condition each output classification without the need for recurrent connections. This allows for parallel training, which considerably speeds up the development process. In the present study, the WaveNet time series module is the same that was utilized in out initial study [11]. It uses four residual blocks with dilation sizes of [1, 2, 4, 8] with filter size 5, reaching a receptive field of 60 frames (69.2 s).

### 4.6. Data Augmentation Methods

Data augmentation methods apply perturbations and/or transformations to the training data so that the actual raw input data values change according to potential intraclass variation observed beyond the training data and without affecting the label identity of the chosen frame. Augmentation aims at either increasing (1) absolute model performance or (2) model robustness in expected real-world settings. We aim to investigate both of these effects with a number of augmentation methods (with modifications) that were previously proposed in [19].

#### 4.6.1. Dropout

Dropout is generally a popular deep learning method to increase model generalization: During training, a binary dropout mask is applied to a given layer, which sets a given percentage of its activation values to zero. Effectively, this forces the neural network to learn robust distributed representations for the same input. For the present study, we experimented with dropout at two different positions: (1) in the input tensor, and (2) in the bottleneck features between the encoder and time series modeling modules. The first case can be thought of data augmentation for the case of packet loss that might occur during a real-world recording. The dropout probability for both cases is set to 30% in the present experiments.

#### 4.6.2. Sensor Dropout

Sensor dropout aims to increase model robustness by randomly dropping one IMU sensor of a minibatch during training (i.e., the sensor’s accelerometer and gyroscope information is set to zero). The probability of a sensor to drop is set to 30%, and the sensor to be dropped is selected randomly from a uniform distribution. This forces the classifier to learn representations that are not dependent on any one particular limb’s movement information. The results in [11] suggest that a combination of three, or even two, of the four sensors is capable of producing nearly on-par classification performance compared to the full set when using an SVM classifier.

#### 4.6.3. Sensor Rotation

Even though the sensor mounts of the Maiju jumpsuit are placed in fixed positions, the actually realized positioning of each sensor with relation to the infant’s body is affected by the body type of the infant. This effect can be simulated with artificial sensor rotation, where a 3 × 3 rotation matrix is used to mix the channels of each sensor. To perform random rotations to the sensors, we utilize the Euler rotation formula, where the yaw (α), pitch (β), and roll (γ) angles are randomly sampled from a uniform distribution between $\pm 10^{\circ }$:(1)$$\begin{array}{c} R_{z}\left(\alpha \right)=\left[\begin{array}{ccc} \cos\alpha & -\sin\alpha & 0\\ \sin\alpha & \cos\alpha & 0\\ 0 & 0 & 1 \end{array}\right] \end{array}$$
(2)$$\begin{array}{c} R_{y}\left(\beta \right)=\left[\begin{array}{ccc} \cos\beta & 0 & \sin\beta \\ 0 & 1 & 0\\ -\sin\beta & 0 & \cos\beta \end{array}\right] \end{array}$$
(3)$$\begin{array}{c} R_{x}\left(\gamma \right)=\left[\begin{array}{ccc} 1 & 0 & 0\\ 0 & \cos\gamma & -\sin\gamma \\ 0 & \sin\gamma & \cos\gamma \end{array}\right] \end{array}$$
(4)$$\begin{array}{c} R\left(\alpha ,\beta ,\gamma \right)=R_{z}\left(\alpha \right)R_{y}\left(\beta \right)R_{x}\left(\gamma \right), \end{array}$$
and the final augmented sensor signal is obtained by multiplying the triaxial accelerometer and gyroscope signals with the resultant matrix.

#### 4.6.4. Time Warping

Time warping augmentation aims to add variability to the training data by altering the speed at which movements occur. This is simulated frame-by-frame by synthesizing a new timestamp differential base vector with the formula
(5)$$dt_{\mathrm{new}}\left(dt_{\mathrm{old}}\right)=2+A\sin\left(2\pi \omega dt_{\mathrm{old}}+2\pi \phi \right),$$
where the frequency (ω), phase (ϕ), and amplitude (*A*) are randomly sampled from a uniform distribution between [0, 1]. The synthesized timebase differential vector ($dt_{\mathrm{new}}$) is then integrated, and each frame’s signals are linearly interpolated to the new timebase. Effectively, this adds sinusoidal fluctuation to the speed of movement while preserving the overall length of each frame. It is to be noted that this method of time warping augmentation does not precisely transform the recorded accelerometer or gyroscope signals into forms that they would take if the actual limb movements would be altered with the same time warping operation.

## 5. Experiments and Results

The goals of the experiments were threefold: (1) to investigate the performance of different sensor encoder and time series modules, (2) to compare their performance with their computational and memory loads, and (3) to study the effects of data augmentation on system performance. The overall aim was to identify the best-performing solution for infant motility assessment using Maiju with reasonable computational requirements.

### 5.1. Training and Evaluation Details

For all of the experiments, each neural network combination was trained end-to-end with random initialization without any pretraining. To mitigate the effect of result variability due to the random initialization, each experiment was repeated three times and their average is reported as the result. The neural network training was performed with stochastic gradient descent using minibatches of 100 frames. The Adam algorithm [21] with parameter values of learning rate = 10 $\times \;10^{-4}$, $\beta_{1}=0.9$ and $\beta_{2}=0.999$ was used to perform the weight updates. Sevenfold cross-validation was used to obtain classification results for the entire dataset. To monitor training, 20% of the minibatches from each recording of the training set were assigned for validation to monitor convergence and to trigger early stopping. The training was performed for a maximum of 250 epochs with an early stopping patience of 30 epochs.

The unweighted average F1-score (UWAF) was chosen as the main performance metric in the present study. The F1-score of one category is the harmonic mean of the precision and recall of that category:(6)$$F1=\frac{2P_{c}R_{c}}{P_{c}+R_{c}}=\frac{tp}{tp+0.5(fp+fn)},$$
where $P_{c}$ is the precision of category *c*, $R_{c}$ is the recall of category *c*, tp is the number of true positive classifications, fp the number of false positives, and fn the number of false negatives. After the F1-score is computed individually for each output category of the dataset, the UWAF can be computed as their average. UWAF is preferable over the standard F1-score with heavily skewed class distributions, as in the case of the present study.

### 5.2. Experiment 1: Comparison of Architecture Models

The first experiment compared the classifier performance of all of the encoder–time series module combinations presented in Section 4.4 and Section 4.5 as measured by UWAF. The results, sorted in ascending order of performance and grouped according to the encoder modules, are presented in Figure 5. In addition, the grouped performances of the individual modules are presented in Figure 6, where statistically significant differences between the architectures are reported with the Wilcoxon rank sum test.

The results show that the best overall performance was achieved with the Conv2D-SI encoder with the WaveNet time series model. The choice of the encoder module is seen to have the largest impact on the classifier performance: the Conv2D-based encoders significantly outperform the competitors. Within the alternative Conv2D models, the simplest model, Conv2D-I, has the weakest performance. The Conv2D-SI model slightly outperforms the previously proposed Conv2D-IS model. The Conv2D-SI model also has the smallest performance variation across the various time series models, which suggests that the bottleneck representations learned by the encoder are the most robust (i.e., the time series module does not need to learn to compensate the shortcomings of the encoder representation).

The choice of the time series model is seen to have a clear hierarchy in performance, where the WaveNet-based model systematically has the best performance. An interpretation for this effect is that the WaveNet model has more powerful modeling capabilities that learn to compensate some of the inefficiencies of inferior encoder modules. Surprisingly, the GRU module systematically outperforms the BGRU module, albeit with a minor margin.

An interesting effect is also seen within the LSTM model: the worse the encoder performance is (i.e., the weaker the bottleneck representation is), the worse the LSTM also performs. With the best-performing encoder (Conv2D-SI), LSTM performance becomes on par with the rest of the RNN-based models. This finding is in line with the conventional wisdom regarding LSTM training instability, and suggests that RNN training might benefit from pretrained bottleneck features that would make the end-to-end training more stable.

### 5.3. Experiment 2: Effects of Model Size and Computational Load to Performance

As further analysis of the first experiment systems, we explored the relationship of model size and model training time to classifier performance. Typically, smaller models have better generalization with a loss to detail of modeling capabilities, whereas larger models have better modeling capacity but are prone to overfitting during training. For research and development purposes, model training time is another feature of interest, as it determines a limit on the rate of experiments that can be performed given a limited number of computational resources. RNN-based systems are known to have considerably slower training times because the error backpropagation through time requires unraveling of the entire computational graph over all of the time steps of the minibatch [22]. In contrast, network structures that do not include recurrent connections can be trained in parallel, which significantly speeds up the training.

The performance—model size—training time comparison of the systems is presented in the bubble chart of Figure 7. The x-axis represents the number of trainable parameters in each model, the y-axis represents the classifier performance, and the bubble areas are proportional to the training times of the given model. The results show that the conventional wisdom regarding dense networks and CNNs holds true within the encoder models: the Conv1D encoder outperforms the dense encoder with almost a 10-fold reduction in trainable parameters. However, the Conv2D encoders clearly exhibit an optimal middle ground where more nuanced parameter sharing with a moderate increase in trainable parameters results in superior classifier performance over the Conv1D and dense encoders.

Within the time series models, the RNN-based modules exhibit considerably longer training times, and the dense and WaveNet time series module training times were roughly equivalent. From the fastest to slowest training times, the ratios are 1 (dense):1 (WaveNet):4.9 (GRU):6.5 (LSTM):7.6 (BGRU). The mean ratio between RNN and non-RNN training times is approximately 6:1. For the encoder models, the training times have ratios of 1 (dense):1.3 (Conv1D):1.4 (Conv2D-I):1.7 (Conv2D-IS):2.1 (Conv2D-SI).

For the final model size experiment, we chose the best-performing system, Conv2D-SI + WaveNet, and tested its performance with a varying number of bottleneck features. As the Conv2D-SI encoder produces sensor-independent features, a decrease in bottleneck layer dimensionality directly reduces the number of features per sensor, thus also reducing the network size. This is an interesting avenue of investigation when considering embedded solutions that would perform the bottleneck feature extraction inside the sensor. Implementing such a scheme would allow data collection without online streaming of the raw data, which would significantly increase recording reliability and sensor battery life. The effect of varying bottleneck size to Conv2D-IS + WaveNet system performance is presented in Figure 8. The results show that halving the bottleneck size to 20 features per sensor (80 total) reduces the number of trainable parameters by 62% and the training time by 23% while having only a 0.5% point drop in performance. This corresponds to a substantial compression ratio of 36:1 compared to the raw sensory data. Further halving of the bottleneck size produces a more considerable drop in performance.

### 5.4. Experiment 3: Model Robustness to Noise

The final set of experiments explored classifier model robustness to noise as well as the effect of various data augmentation methods. For these experiments, we chose the best-performing system, Conv2D-IS + WaveNet, alongside two of the systems with the lowest number of trainable parameters: Conv1D + dense and Conv1D + GRU. Robustness was tested for sensor-dropout and packet-loss cases under two conditions: moderate and severe.

For sensor dropout, one (moderate condition) or two sensors (severe condition) were dropped for each recording. Every possible combination of the dropped sensors was tested, and the average performance across the tests is reported in the results. For packet loss, the moderate condition simulated 25% random dropout in four sample bursts per sensor. The severe condition used a dropout value of 50%. For both cases, the dropout simulation was repeated for five times, and the average result is reported.

The results are presented in Figure 9. For all of the systems, the application of data augmentation produces minor differences compared to the baseline without augmentation when tested with clean data. For the sensor dropout conditions, considerable boosts to performance can be observed with the augmentation methods where sensor dropout is present: an average effect of +5.5% points for one sensor dropped and +12.5% points for the two sensors dropped. For the Conv1D-based models, input-level dropout also produces a considerable performance boost (average effect of +5% points and +7% points).

The results for the packet-loss case are somewhat more diverse: for the moderate case, the performance of the Conv2D-SI + Wavenet model is on par with the “no dropout” case, even without augmentation. This further suggests that the weight sharing in the Conv2D-SI encoder produces a robust low-dimensional representation for the movement sensor data. Rather surprisingly, input-level dropout does not increase performance of this system even in the severe packet loss case, but an increase in performance is observed with sensor dropout during training. For the Conv1D encoder-based systems, input-level dropout produces the greatest augmentation effect for the moderate and severe packet loss cases, whereas sensor dropout produces a smaller positive effect. In summary, the combination of input-level dropout with sensor dropout seems to be the most beneficial training strategy for the Maiju classifier, as it maintains the clean baseline performance while making the classifier considerably more robust in adverse conditions.

## 6. Discussion

This study shows that the choice of end-to-end neural network architectures has a major impact on the performance of human activity recognition from multisensor recordings. The choice of the encoder module (responsible for intra- and/or intersensor channel fusion) has the greatest significance in the resulting performance. Out of the tested encoders, the best performance was obtained with the “Conv2D-SI” model which uses parallel two-dimensional convolutions for intrasensor channel fusion with shared weights for all sensors. Furthermore, experiments regarding the bottleneck size of the the “Conv2D-SI” encoder indicate that a relatively compact feature representation is obtainable for within-sensor feature extraction without a drastic loss in classifier performance. Comparison of time series models reveals several key effects. First, the best-performing system, the “WaveNet” architecture based on feedforward dilated convolutions, outperforms the RNN-based models in performance, training time, and training stability, making it the preferred time series modeling method for the task. The same results also show the importance of input-feature robustness when training RNN-based models, especially LSTMs. This suggests that some level of pretraining (supervised or unsupervised) would be preferable before inserting an RNN module into an end-to-end classifier architecture.

The experiments comparing data augmentation methods did not find additional benefit to classifier performance when using clean data as input. However, when studying model robustness in noisy conditions, such as simulated packet loss or dropped sensor scenarios, dropout-based (input dropout or sensor dropout) augmentation provided a considerable boost to performance without affecting the clean data performance. We speculate that the lack of performance boost from data augmentation is due to performance ceiling caused by the ambiguity in defining the gold-standard annotations of infant movement categories before the infants reach archetypal adult-like motor skills. Our previous study showed that the classifier output reaches the inter-rater agreement κ valuebetween three independent annotators [11].

In conclusion, this study suggests how to optimize end-to-end neural network training for multichannel movement sensor data using moderate-sized datasets. First, the design of the encoder/channel fusion component of the network greatly benefits from weight sharing and appropriate handling of data modalities from different sources. Second, feedforward convolutional approaches with wide receptive fields are well suited for time series modeling, and outperform RNNs in terms of accuracy, training speed, and robustness on our data. Finally, training data inflation by means of data augmentation is, at best, a secondary avenue in increasing classifier performance compared to the overall network structure and dataset design. Regarding the Maiju jumpsuit, our future research interests include the expansion of the training dataset to contain infants up to 14 months in age, and in exploring the effect of unsupervised representation learning techniques such as contrastive predictive coding [23] to the classification results.

## Figures and Tables

**Figure 1 sensors-23-03773-f001:**
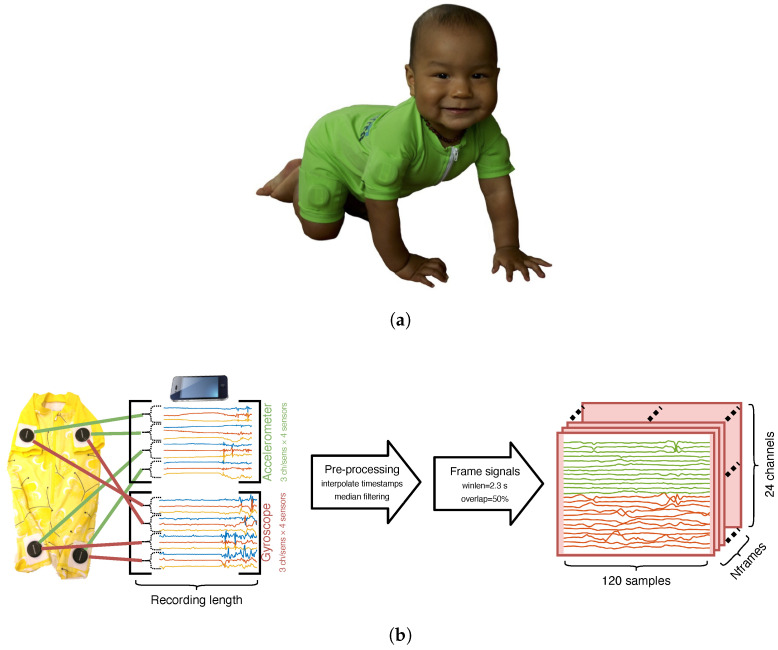
The Maiju jumpsuit. The picture in (**a**) is published with parental consent. (**a**) The Maiju jumpsuit worn by a 10-month-old infant. (**b**) Illustration of the recording setup and data preprocessing steps. The green and red lines from the sensors denote the accelerometer and gyroscope data, respectively.

**Figure 2 sensors-23-03773-f002:**
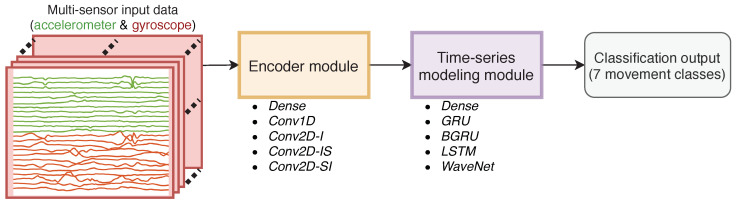
Overall schematic of the compared classifier architectures.

**Figure 3 sensors-23-03773-f003:**
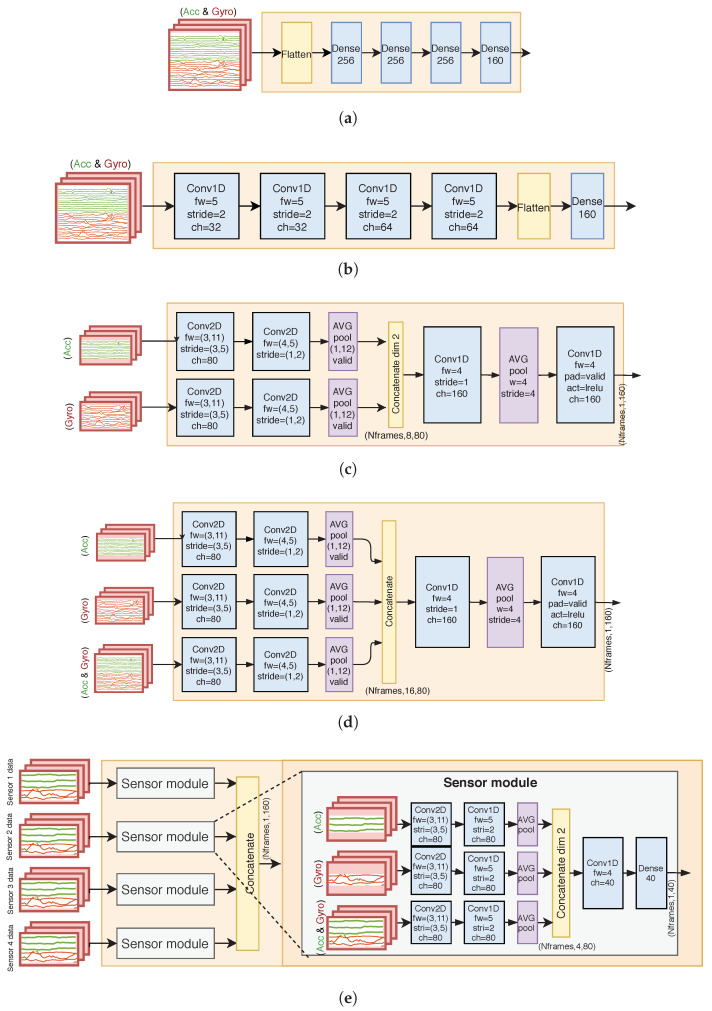
Compared encoder module architectures. (**a**) Dense encoder; (**b**) 1-D CNN (Conv1D) encoder; (**c**) 2-D CNN with individual paths (Conv2D-I) encoder; (**d**) 2-D CNN with individual and shared paths (Conv2D-IS) encoder; (**e**) 2-D CNN with sensor-independent modules (Conv2D-SI) encoder.

**Figure 4 sensors-23-03773-f004:**
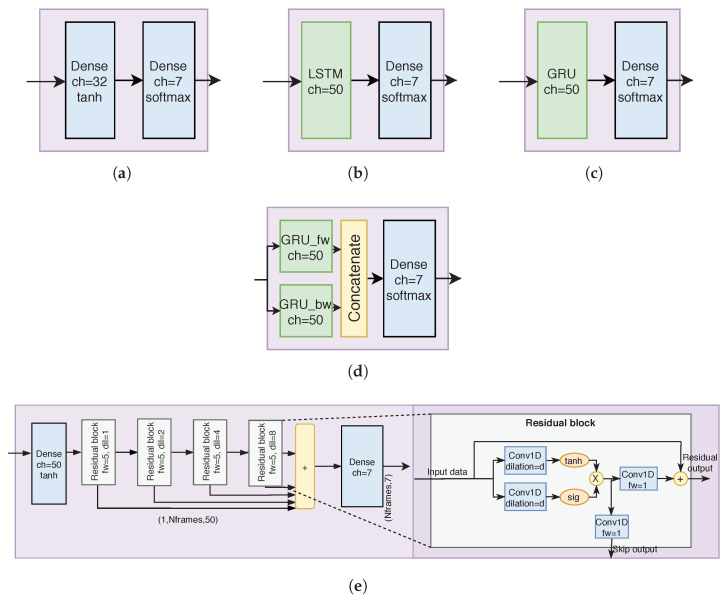
Compared time series module architectures. (**a**) Dense time series model; (**b**) LSTM time series model; (**c**) GRU time series model; (**d**) BGRU time series model; (**e**) WaveNet time series model.

**Figure 5 sensors-23-03773-f005:**
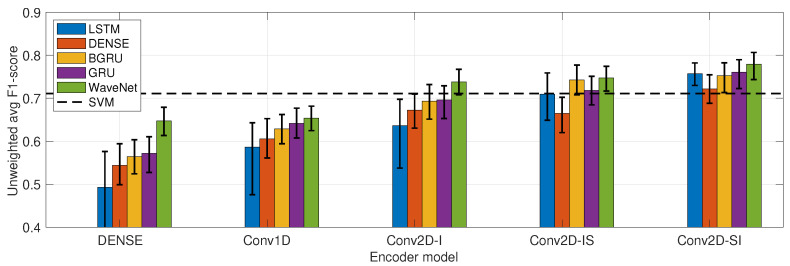
Unweighted average *F*1-scores as a function of encoder modules (x-axis placement) and time series modules (color coding). The 95% confidence intervals obtained with the bootstrap method (record-level sampling, 10,000 iterations) are presented as whiskers.

**Figure 6 sensors-23-03773-f006:**
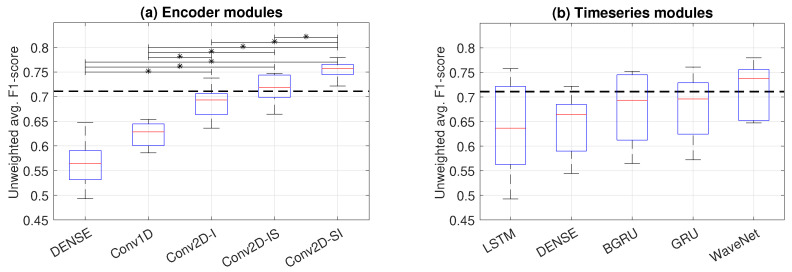
Pooled performance of encoder modules (**a**) and time series modules (**b**) as boxplots showcasing their range, interquartile range, and median. Statistically significant differences (Wilcoxon rank sum test; p<0.05) between the average effect of the modules are marked with ‘*’.

**Figure 7 sensors-23-03773-f007:**
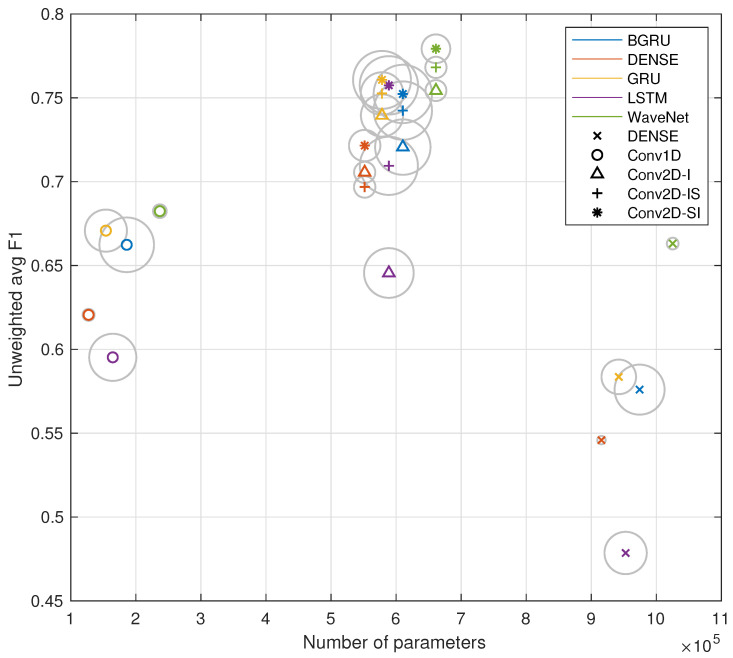
Average performance of trained systems as a function of model size (x-axis) and the generation time of the test set (symbol area).

**Figure 8 sensors-23-03773-f008:**
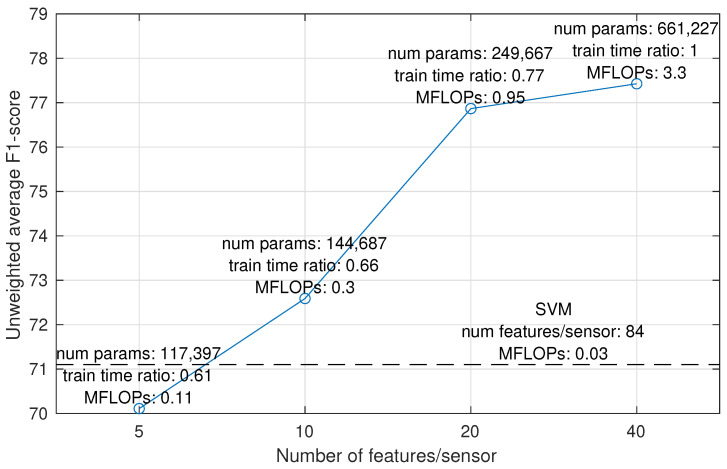
Effect of the number of bottleneck features for Conv2D-SI to the unweighted average F1-score. The performance of the feature + SVM model is shown as a reference (dashed line). The computational complexity is expressed as the number of MFLOPs (mega floating point operations) required to perform the feature extraction within each sensor for a single frame of data.

**Figure 9 sensors-23-03773-f009:**
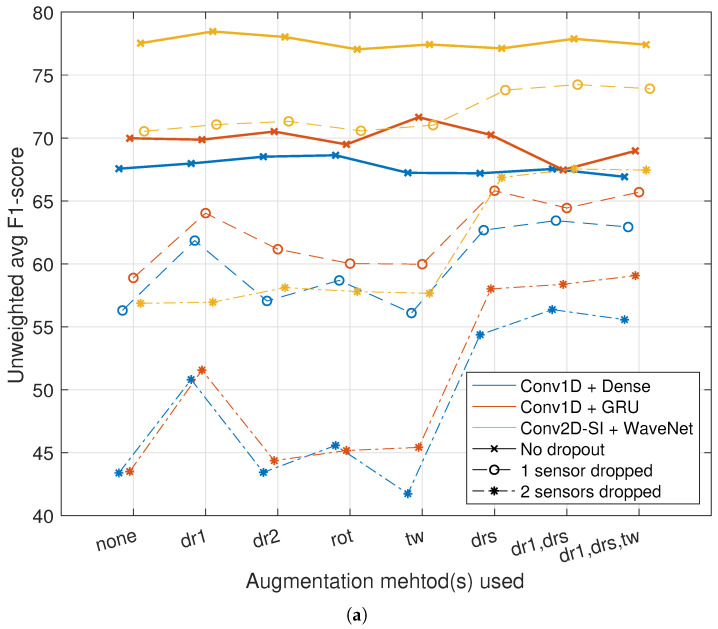
Classifier model robustness to noise and the effect of data augmentation. Augmentation method key: dr1 = input dropout; dr2 = bottleneck dropout; rot = rotation; tw = time warping; drs = sensor dropout. (**a**) Sensor dropout results. (**b**) Packet loss results.

**Table 1 sensors-23-03773-t001:** Details of the utilized infant motility dataset.

	Overall Statistics
Number of recordings	22
Participant age (months)	Avg. 6.6 ±0.84 (min 4.5, max 7.7)
Recording length (min)	Avg. 28.9 ±7.7 (min 8.6, max 40.4)
Total length	10 h 35 min (33,021 frames)
**Movement Categories**	**Total Length (Min, Frames, % Present)**
Still	376 min, 19,547 frames, 63.5%
Proto movement	182 min, 9483 frames, 28.7%
Turn L	9 min, 485 frames, 1.5%
Turn R	9 min, 466 frames, 1.5%
Pivot L	18 min, 978 frames, 3.0%
Pivot R	20 min, 1030 frames, 3.1%
Crawl commando	20 min, 1032 frames, 3.1%

## Data Availability

The data or materials for the experiments reported here can be made available at reasonable request and within relevant legal constraints.

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
