# Peer review of "Comparison of End-to-End Neural Network Architectures and Data Augmentation Methods for Automatic Infant Motility Assessment Using Wearable Sensors"

_sensors, 2023, doi:10.3390/s23073773_

Round 1

Reviewer 1 Report

the manuscript is reviewed. the comments are

author should include quantitative data in the abstract 

check spelling in figure 9 caption "Method".

Details about the experimental hardware is missing.

 1. What is the main question addressed by the research?
> author has developed wearable sensors and applied neural network architectures for automatic infant motility assessment.
2. Does it address a specific gap in the field?
> yes.
3. What does it add to the subject area compared with other published
material?
> the author has developed the NN. they have published the similar papers.
4. What specific improvements should the authors consider regarding the
methodology? What further controls should be considered?
> hardware part is missing.
5. Are the conclusions consistent with the evidence and arguments presented
and do they address the main question posed?
> yes
6. Are the references appropriate?
> yes
7. Any additional comments on the tables and figures?
> include standard deviation in fig 5

Author Response

  1. What specific improvements should the authors consider regarding the
    methodology? What further controls should be considered?
    > hardware part is missing.

We have added a more detailed description about the sensor hardware into Section 3: Data Collection Method.

  1. Any additional comments on the tables and figures?
    > include standard deviation in fig 5

We have added 95% confidence intervals to the values, obtained by the bootstrap method by randomly sampling the recordings.

Reviewer 2 Report

The authors have developed various neural network architectures for assessing infant motility. The paper is potentially suitable for publication, but minor revisions are necessary. Here are some suggestions for the authors:

The paper utilizes raw signals as input and employs neural networks for feature extraction and final classification. It would be beneficial for the authors to compare these deep learning approaches with traditional logistic regression models. Specifically, a comparison with state-of-the-art logistic regression models that use handcrafted features would be informative. Such a comparison could provide valuable insights into the necessity of using neural networks in this context.

Given that these algorithms may ultimately be implemented in embedded systems, computational load is an essential consideration. The authors should compare the computational load of the proposed neural network architectures and evaluate them against logistic regression models that use handcrafted features. This comparison will help readers understand the practicality of these methods in real-world applications.

Author Response

Q1. The paper utilizes raw signals as input and employs neural networks for feature extraction and final classification. It would be beneficial for the authors to compare these deep learning approaches with traditional logistic regression models. Specifically, a comparison with state-of-the-art logistic regression models that use handcrafted features would be informative. Such a comparison could provide valuable insights into the necessity of using neural networks in this context.

A1. We have added a baseline result to Figures 5, 6, and 8 to showcase the performance with the Feature + ensemble SVM classifier reported in our previous study.

Q2. Given that these algorithms may ultimately be implemented in embedded systems, computational load is an essential consideration. The authors should compare the computational load of the proposed neural network architectures and evaluate them against logistic regression models that use handcrafted features. This comparison will help readers understand the practicality of these methods in real-world applications.

A2. Since we have a multi-sensor setup, it does not make sense to compare full classifier stacks in terms of their computational load: The data from each sensor needs to be gathered into a single device which does the final classification, which will not have strict restrictions. A suitable way to do this experiment would be to compare the computational load of the feature computation against the sensor-specific first layers of a neural network that could be implemented within the embedded system. We have added information about the number of floating point operations (Mega FLOPs) in each system of Figure 8, as well as a line depicting the SVM performance.

Reviewer 3 Report

This work investigates the use of different end-to-end neural network architectures for processing infant motility data from wearable sensors. The choice of end-to-end neural network architectures has a major impact on the performance of human activity recognition from multi-sensor recordings and the choice of the encoder module has the greatest significance in the resulting performance. The manuscript is well written and data presented is reasonable, so I think it can be published after minor revisions as below:

1.   In the first paragraph of Introduction, some related references should be added, such as “Faisal A I, et al. Monitoring methods of human body joints: State-of-the-art and research challenges[J]. Sensors, 2019, 19(11): 2629.” and “Hong Y, et al. Highly anisotropic and flexible piezoceramic kirigami for preventing joint disorders[J]. Science Advances, 2021, 7(11): eabf0795.”

2.  The full spelling of an abbreviation is required when it first appears. Please check it through the papers, such as the “RNN” in the abstract.

Author Response

 Q1. In the first paragraph of Introduction, some related references should be added, such as “Faisal A I, et al. Monitoring methods of human body joints: State-of-the-art and research challenges[J]. Sensors, 2019, 19(11): 2629.” and “Hong Y, et al. Highly anisotropic and flexible piezoceramic kirigami for preventing joint disorders[J]. Science Advances, 2021, 7(11): eabf0795.”

A2. We have added more recent citations to the manuscript:

Franchak et al. 2021 - A Contactless Method for Measuring Full-Day, Naturalistic Motor Behavior Using Wearable Inertial Sensors

Chen et al. 2016 - A Review of Wearable Sensor Systems for Monitoring Body Movements of Neonates

Zhou et al. 2019 - quantifying Caregiver Movement when Measuring Infant Movement across a Full Day: A Case Report

Deng et al. 2020 - How Many Days are Necessary to Represent Typical Daily Leg Movement Behavior for Infants at Risk of Developmental Disabilities?

Yun et al. 2020 - Ultra-Low Power Wearable Infant Sleep Position Sensor

Q2. The full spelling of an abbreviation is required when it first appears. Please check it through the papers, such as the “RNN” in the abstract.

A2. We have double-checked the manuscript to include the full spellings of abbreviations when they first appear.

Reviewer 4 Report

In this work, the authors investigated the use of different end-to-end neural network architectures for processing infant motility data from wearable sensors. The results indicated that the choice of the encoder module had a major impact on classifier performance. The signal- and sensor-dropout-based augmentation strategies provided considerable boosts to performance without negatively affecting the baseline performance. This work may benefit other HAR applications that are based on multi-sensor setups. Overall, the design results can verify the effectiveness of this proposed thought. Some suggestions are as below.

(1) What are the different between this work and others, such as methods and performance? The authors may compare other with necessary description to better explain the innovation of this work.

(2) The text in Figures 6 and 9 is unclear. The authors should improve the readability.

(3) The range of references is somewhat narrow. There is no reference published in the journal. As to prove relevance of the submitted manuscript to the journal, some recent references should be dated most preferably 2021-2022.

(4) How accurate is this wearable sensor? The authors should also provide a picture of the device during test.

Author Response

Q1. What are the different between this work and others, such as methods and performance? The authors may compare other with necessary description to better explain the innovation of this work.

A1. To the best of our our knowledge, infant movement classification has not been studied at the present level of detail in other studies. Very similar results for posture classification have been recently reported on Franchak et al. 2021 - A Contactless Method for Measuring Full-Day, Naturalistic Motor Behavior Using Wearable Inertial Sensors, which has now been referenced in the revised manuscript.

Q2. The text in Figures 6 and 9 is unclear. The authors should improve the readability.

A2. The font size has been increased in Figure 6. The sizes of Figures 9a,b have been increased.

Q3. The range of references is somewhat narrow. There is no reference published in the journal. As to prove relevance of the submitted manuscript to the journal, some recent references should be dated most preferably 2021-2022.

A3. We have added more recent citations to the manuscript:

Franchak et al. 2021 - A Contactless Method for Measuring Full-Day, Naturalistic Motor Behavior Using Wearable Inertial Sensors

Chen et al. 2016 - A Review of Wearable Sensor Systems for Monitoring Body Movements of Neonates

Zhou et al. 2019 - quantifying Caregiver Movement when Measuring Infant Movement across a Full Day: A Case Report

Deng et al. 2020 - How Many Days are Necessary to Represent Typical Daily Leg Movement Behavior for Infants at Risk of Developmental Disabilities?

Yun et al. 2020 - Ultra-Low Power Wearable Infant Sleep Position Sensor

Q4. How accurate is this wearable sensor? The authors should also provide a picture of the device during test.

A4. We have added Figure 1a of the jumpsuit worn by a 10-month-old infant. Sensor details have been added to Section 3: Data collection method.